

# Sugarcane/peanut intercropping system improves physicochemical properties by changing N and P cycling and organic matter turnover in root zone soil

Xiumei Tang[1,2], Yixin Zhang[1], Jing Jiang[1], Xiuzhen Meng[1], Zhipeng Huang[1], Haining Wu[1], Liangqiong He[1], Faqian Xiong[1], Jing Liu[1], Ruichun Zhong[1], Zhuqiang Han[1] and Ronghua Tang[1]

[1] Guangxi Academy of Agricultural Sciences, Cash Crops Research Institute, Nanning, Guangxi, China
[2] Guangxi Academy of Agricultural Sciences, Guangxi Crop Genetic Improvement and Biotechnology Laboratory, Nanning, Guangxi, China

## ABSTRACT

**Background**. The sugarcane/peanut intercropping system is a specific and efficient cropping pattern in South China. Intercropping systems change the bacterial diversity of soils and decrease disease rates. It can not only utilized light, heat, water and land resources efficiently, but also increased yield and economic benefits of farmers.

**Methods**. We determined soil nutrients, enzymes and microbes in sugarcane/peanut intercropping system, and analyzed relevance of the soil physicochemical properties and the genes involved in N and P cycling and organic matter turnover by metagenome sequencing.

**Results**. The results showed that sugarcane/peanut intercropping significantly boosted the content of total nitrogen, available phosphorus, total potassium, organic matter, pH value and bacteria and enhanced the activity of acid phosphatase compared to monocropping. Especially the content of available nitrogen, available phosphorus and organic matter increased significantly by 20.1%, 65.3% and 56.0% in root zone soil of IP2 treatment than monocropping treatment. The content of available potassium and microbial biomass carbon, as well as the activity of catalase, sucrase and protease, significantly decreased in intercropping root zone soil. Intercropping resulted in a significant increase by 7.8%, 16.2% and 23.0% in IS, IP1 and IP2, respectively, of the acid phosphatase content relative to MS. Metagenomic analysis showed that the pathways involved in carbohydrate and amino acid metabolism were dominant and more abundant in intercropping than in monocropping. Moreover, the relative abundances of genes related to N cycling (*glnA*, *GLUD1_2*, *nirK*), P cycling (*phoR*, *phoB*) and organic matter turnover (*PRDX2_4*) were higher in the intercropping soil than in the monocropping soil. The relative abundance of *GLUD1_2* and *phoR* were 25.5% and 13.8% higher in the IP2 treatment respectively, and *bgIX* was higher in IS treatment compared to the monocropping treatment. Genes that were significantly related to phosphorus metabolism and nitrogen metabolism (*TREH*, *katE*, *gudB*) were more abundant in intercropping than in monocropping.

**Conclusion**. The results of this study indicate that the intercropping system changed the numbers of microbes as well as enzymes activities, and subsequently regulate genes

Corresponding authors
Xiumei Tang, tangxi-umei196@163.com
Ronghua Tang, tronghua@163.com

involved in N cycling, P cycling and organic matter turnover. Finally, it leads to the increase of nutrients in root zone soil and improved the soil environment.

## INTRODUCTION

Sugarcane is an important agro-economic sugar crop utilized as a biofuel worldwide and is also one of the primary cash crops in Guangxi Province, China (*Chen et al., 2019*; *Solanki et al., 2016*). Intercropping cultivation systems are attributed to reduced production costs, improved yields (*Kamruzzaman & Hasanuzzaman, 2007*) and suitable utilization of natural resources (*Solanki et al., 2016*), and reduced impacts of pests and diseases (*Cong et al., 2015*; *Boudreau, 2013*; *Damicone et al., 2007*). Plants with different growth habits and growth periods may contribute to optimal and rational use of resources (*Verma et al., 2014*). Sugarcane is a kind of crop with wide row spacing and slow seedling growth (*Shen, Zhao & Chen, 2018*) and is suitable to be intercropped with other crops that grow rapidly, such as peanut and soybean. Intercropping cultivation can efficiently utilize light and nutrients and increase yields (*Li et al., 2010*; *Quan et al., 2013*). Sugarcane/peanut intercropping makes full use of soil nutrients and land resources and increases famers' economic benefits, which contributes to the development of efficient and sustainable production of sugarcane and peanut.

Intercropping affects microbial communities and chemical properties in root zone soil (*Cao et al., 2017*) . The interactions among microbes, nutrients and enzymes in intercropping systems leads to an increase or decrease in microbe quantity and enzyme activity, contributing to the improvement of the soil micro-ecological environment (*Zhang et al., 2012b*; *Zhou et al., 2019*). These interactions affect plant productivity directly or indirectly. Soil microbial communities are involved in various ecosystem processes, including mineralization and mobilization of nutrients required for plant growth (*Regehr et al., 2015*; *Song et al., 2006*), increasing the availability and supply of limiting nutrients (*Bainard et al., 2012*), and improving soil structure (*Tian et al., 2019*). *Shen, Zhao & Chen (2018)* reported that intercropping with peanut and Si application helped to increase the yield and plant height of sugarcane. Previous studies have shown that the sugarcane intercropping system enhanced the diazotrophic population (*Shen et al., 2014*; *Solanki et al., 2016*) and significantly increased the phosphorous content while decreasing the pH of root zone soil compared with monocropping (*Qin et al., 2019*). According to *Tang et al., (2016b)*, high P levels could enhance the advantages of intercropping, thereby affecting root zone microbial properties. On the basis of higher microbial activity, the intercropping system could reduce the cost of application of nitrogen and phosphorus fertilizing. In addition, higher natural biological nitrogen-fixing activity was identified as an important factor contributing to enhancing the yield of sugarcane (*Liu et al., 2019b*).

Several studies have revealed that the activity of soil enzymes, the effective nitrogen and phosphorus contents and the microbe number of root zone soil were significantly increased in sugarcane/soybean intercropping (*Li et al., 2012*; *Peng et al., 2014*; *Solanki et al., 2019*; *Solanki et al., 2018*). The maize/peanut intercropping system enhances strong light utilization ability and leads to higher efficiency of soil nutrients than monoculture planting patterns (*Jiao et al., 2016*; *Wang et al., 2019*; *Zhang et al., 2019a*). Cassava/peanut intercropping is more conducive to transforming cassava root zone soils into high fertility bacteria (*Xu et al., 2016*). Peanuts secrete protons and organic acids to activate insoluble inorganic phosphorus, promoting the absorption of phosphorus in root zone soil, which is conclusively beneficial to the growth of both peanut and cassava (*Lin et al., 2018*; *Liu et al., 2019c*). *Verma et al. (2014)* reported that higher organic C in intercropping system inputs through the decomposition of plant residues helped to increase microbial activities, which enhanced plant growth.

The microecological environment plays an important role in the growth of intercropped crops and the development of sustainable agriculture. Although the impact of sugarcane/peanut intercropping on soil nutrients, soil enzyme activity and bacterial population has been investigated in several studies (*Li et al., 2012*; *Liu et al., 2019b*; *Qin et al., 2019*; *Shen et al., 2014*; *Shen, Zhao & Chen, 2018*), little is known about how sugarcane/peanut intercropping affects the microecological environment, especially the interaction mechanism of microbes-nutrients-enzymes involved in N/P cycling and organic matter turnover in intercropping systems. In this study, our objective was to investigate the nutrients, root zone soil microbes and enzyme activity under sugarcane/peanut intercropping conditions, including analysis of N, P and K content, organic matter content, pH value, microbe quantity and soil metagenomic sequencing. Through metagenomic sequencing, we can not only obtain the characteristic information of all the microbial communities in the sample but also perform the analysis of genes and metabolic pathways (*Zhang et al., 2019b*). This study provides comparative metagenomic insights for evaluating the impacts of sugarcane/peanut intercropping on the microecological environment.

## MATERIALS AND METHODS

### Experimental site and plant materials

The experiments were performed at the Wuxuan Demonstration Base (23°50′84″N, 109°53′81″E), Luxin town, Laibin city, Guangxi Province, China. The field site was previously used for monocropping sugarcane. The monocropping sugarcane was planted and managed in a conventional manner based on local farmers' methods. The tested soil was sandy soil , in which organic matter content, total nitrogen content, total phosphorus content, total potassium content, available nitrogen content, available phosphorus content and available potassium content were 18.280 g/kg, 1.022 g/kg, 0.315 g/kg, 6.583 g/kg, 82.83 mg/kg, 120.78 mg/kg and111.67 mg/kg, respectively. The pH value was 7.02. The sugarcane variety "Guitang42" and the shade-tolerant peanut variety "Guihua 836" were provided by the Cash Crops Research Institute of the Guangxi Academy of Agricultural Sciences.

## Experimental design and fertilization management

The field site was previously used for monocropping sugarcane. This is double-season work and we planted peanut for two seasons in the same experimental plots with sugarcane from 2018 to 2019. The growth period of peanut is about 4 months and the growth period of sugarcane is about 9 months. We sowed peanut in March, 2018 and March, 2019 respectively. We planted sugarcane once in 2018 and the stubble cane grew in 2019. After the harvest of sugarcane, the bud left by the old sugarcane in soil sprouted unearthed under appropriate environmental conditions (temperature and humidity) and grew into new sugarcane, which was called the stubble cane. The planting time, order and management in detail are as follows:

On March 10th, 2018, sugarcane and peanut were planted simultaneously in the field. The field site was previously used for monocropping sugarcane. On July 10th and December 28th, 2018, peanut and the sugarcane were harvested respectively. On March 10th, 2019, peanut was intercropped with stubble cane. Monocropping sugarcane (MS) was the control, and the sugarcane/peanut intercropping system was the treatment group, which contained intercropping sugarcane (IS), intercropping peanut in the edge row (near the sugarcane) (IP1) and intercropping peanut in the middle row (far away from the sugarcane) (IP2) (Fig. 1). The soils in the roots of MS crops were compared with those of IS, IP1 and IP2 crops in the sugarcane/peanut intercropping system. For MS, sugarcane was planted with a row spacing of 1.2 m. For IS and IP, three lines of peanut were planted next to one line of sugarcane. The line spacing between sugarcane and peanut was 0.8 m. The line spacing for sugarcane was 2.4 m, and that for the intercropped peanuts was 0.4 m (Fig. 1). The experiment was arranged in plots (8 m × 10 m) in a randomized design with three replicates in each treatment. The fertilization regimes applied to different crops depended on actual amount of fertilizer required, and peanut required fertilizer less than sugarcane. All peanut treatments received 450 kg ha$^{-1}$ compound NPK granulated fertilizers (N-P$_2$O$_5$-K$_2$O = 15-15-15) and 750 kg ha$^{-1}$ fused calcium-magnesium phosphate fertilizer (available P$_2$O$_5$ 18%). All sugarcane treatments only received 750 kg ha$^{-1}$ compound NPK granulated fertilizers. The crops were irrigated two times during crop growth based on crop water requirements and soil water content. Pesticides and herbicides were applied approximately two months after sowing. On July 10th, 2018, peanut were harvested. After harvest, the residues of peanut covered between the lines of sugarcane in order to moisturize and enrich the soil. Then we added 1,500 kg ha-1 compound NPK granulated fertilizers and the sugarcane continued to grow until the harvest in December 28th, 2018. We cut the stalk of sugarcane and its root remained in soil, which will grew into the stubble cane next year.

On March 10th, 2019, peanut was planted in plots where the sugarcane was planted the former year . So this year the peanut was intercropped with stubble cane. After the harvest of sugarcane, the bud left by the old sugarcane in soil sprouted unearthed under appropriate environmental conditions (temperature and humidity) and grew into new sugarcane, which was called the stubble cane. The area of plots, fertilization and management were same as mentioned in 2018. In July 8th, 2019, we harvested peanut and collected soil samples for analysis. The soils in the roots of MS crops were compared with those of IS,
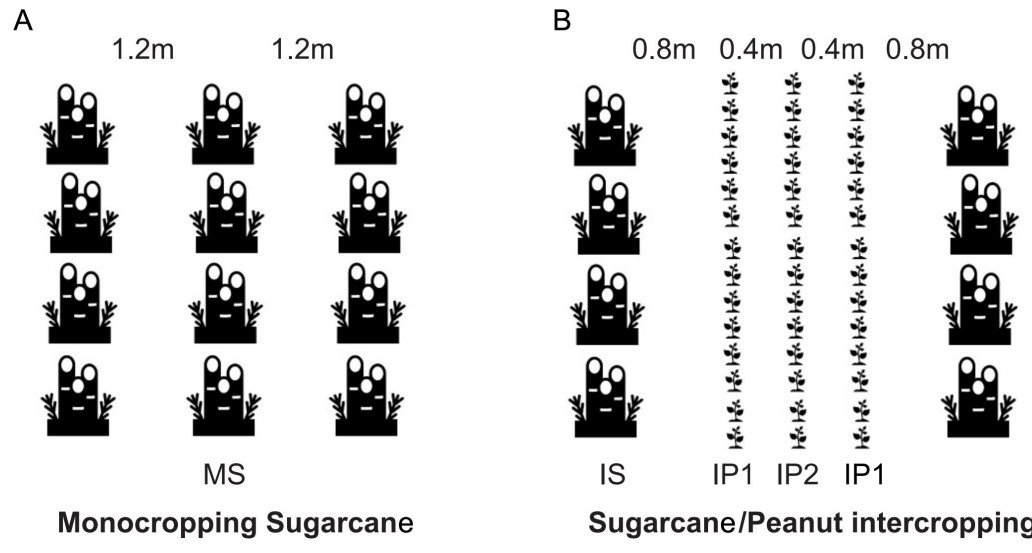

**Figure 1** (A) Monocropping sugarcane (MS); (B) intercropping sugarcane (IS), intercropping peanut in the first line (IP1) and intercropping peanut in the second line (IP2).

IP1 and IP2 crops in the sugarcane/peanut,intercropping system. The sugarcane continued to grow until the harvest in December 26th, 2019.

## Soil sampling

On July 25th, 2019, the time to harvest the mature peanuts, ten plants of sugarcane and peanut per treatment were uprooted. The soil from both bulk soil and soil attached to the plant roots was collected, mixed and separated into three sealed virus-free bags for subsequent assays (*Dragana et al., 2017*). One bag (about 100 g) was kept in the refrigerator at 4 °C and used for culturable soil microbe determination. One bag (about 50 g) was stored in a refrigerator at −80 °C and used to extract soil DNA and for metagenome sequencing. The last bag (about 400 g) was dried naturally, ground and sieved and used for the determination of the nutrient content and the soil enzyme activity.

## Soil physicochemical property analysis

The physicochemical properties were measured according to previous reports (*Zhang et al., 2018*). The available N, available P, available K and organic matter contents were measured by the alkaline hydrolysis diffusion method (*Page, Miller & Keeney, 1982*), sodium bicarbonate extraction/Mo-Sb colorimetry (*Colwell, 1963*), ammonium acetate extraction/flame photometry using flame spectrophotometer (FP6410,China) (*Guo, 2014*) and the potassium dichromate titrimetric method (*Wang et al., 2014*), respectively. Catalase activityand sucrase activity (*Mueller, Riedel & Stremmel, 1997*) were measured by permanganate titration, sodium thiosulfate titration. Proteinase activity, urease activity (*Cordero, Snell & Bardgett, 2019*) and acid phosphatase activity (*Li et al., 2004*) were measured by ninhydrin colorimetry, indophenol blue colorimetry and the disodium

phosphate benzene colorimetric method using ultraviolet visible photometer (UV-1750, Japan), respectively.

## Determination of soil microbial abundance

Soil microbial abundance was measured by the conventional microculture method. Bacteria were cultured in beef extract-peptone medium, and fungi were cultured in Martin medium, and actinomycetes were cultured in Gao 1 medium using constant temperature incubator (DH3600B, China). The microbial biomass carbon (MBC) and nitrogen (MBN) were determined by the chloroform fumigation-$K_2SO_4$ extraction method, and the microbial biomass of soil (MBP) was determined by the fumigation-$NaHCO_3$ extraction method (*Wu et al., 2006*).

## Genomics DNA extraction

The soil samples of each treatment were put in a blender to be fully smashed and homogenized, of which 0.2 g were applied to DNA extracting. Total genomic DNA was extracted from soil samples using the E.Z.N.A.® Soil DNA Kit (Omega Bio-tek, Norcross, GA, U.S.) according to manufacturer's instructions. The microbial community DNA was extracted using a NucleoSpin Soil Kit (Macherey-Nagel, Germany) following the manufacturer's instructions. DNA was quantified with a Qubit Fluorometer by using a Qubit dsDNA BR Assay kit (Invitrogen), and the quality was checked by running aliquots on a 1% agarose gel.

## Library construction and sequencing

DNA extract was fragmented to an average size of about 400 bp using Covaris M220 (Gene Company Limited, China) for paired-end library construction. Paired-end library was constructed using NEXTFLEX Rapid DNA-Seq (Bioo Scientific, Austin, TX, USA). Adapters containing the full complement of sequencing primer hybridization sites were ligated to the blunt-end of fragments. The selected fragments were subjected to end repair, 3ʹadenylation, adapter ligation, and PCR amplification, and the products were purified by magnetic beads. The double stranded PCR products were heat-denatured and circularized by the splint oligo sequence. The single strand circle DNA (ssCir DNA) was formatted as the final library and qualified by QC. The qualified libraries were sequenced on the MGISEQ-2000 platform (BGI-Shenzhen, China) (*Zhu et al., 2020*; *Yang et al., 2020*).

## Statistical analysis

All the raw data were trimmed by SOAPnuke v.1.5.2 (*Chen et al., 2018b*). High-quality reads were *de novo* assembled using Megahit (*Li et al., 2015*) software. Assembled contigs with lengths less than 300 bp were discarded in the following analysis. Genes were predicted over contigs by using MetaGeneMarker (2.10) (*Zhu, Lomsadze & Borodovsky, 2010*). Redundant genes were removed using CD-HIT (*Fu et al., 2012*) with an identity cutoff of 95%. To generate the taxonomic information, the protein sequences of genes were aligned against the NR database using DIAMOND (*Buchfink, Xie & Huson, 2015*) with an $E$ value cutoff of $1e^{-5}$. Based on the MEGAN (*Huson et al., 2007*) LCA algorithm, taxonomic annotation was assigned. To obtain functional information, the protein sequences were aligned against

the eggNOG database (2015-10), CAZy database (2017-09), COG database (2014-11), Swiss-prot database (2017-07), and CARD database (4.0) by DIAMOND (*Buchfink, Xie & Huson, 2015*) with an $E$ value cutoff of $1e^{-5}$. Data of metabolic pathways, the normality and variance homogeneity, relative abundances of genes and network analysis were measured and analyzed as previously described in *Zheng et al. (2019)*. We analyzed the control capabilities of genes and produced figures based on betweenness centrality scores measured from our data. The taxonomic and functional abundance profiles were analyzed from the reads which were aligned to the genes using Botwie2 (*Langmead & Salzberg, 2012*) with the default setting. Based on the abundance profiles, the features (Genera, Phyla and KOs) with significantly differential abundances across groups were determined using Wilcoxon's rank sum test (*Matsouaka, Singhal & Betensky, 2018*). $P$ values for multiple testing were corrected using the BH (*Yekutieli & Benjamini, 2001*) method, and corrected $P$-values<0.065 were considered to be significant. Differentially enriched KEGG pathways were identified according to reporter scores (*Patil & Nielsen, 2005*). The means and standard errors of the MS, IS, IP1 and IP2 with three replicates were analyzed by one-way variance analysis with SPSS 24.0 (IBM), and S-N-K's test was used to test the homogeneity of variance.

All of the sequence data have been deposited in the NCBI Sequence Read Archive (SRA) database under accession number SRP267937 (SAMN15324604- SAMN15324615). We have submitted the assemblies to GenBank under the accession from JACZCT000000000 to JACZDE000000000 and the contigs under accession PRJNA640507.

## RESULTS

### Effects of sugarcane/peanut intercropping on the physicochemical properties of soil

The soil nutrients of the root zone soil in the different treatments are given in Table 1. Compared with sugarcane (MS), the available phosphorus content was significantly higher in intercropping treatments. Intercropping sugarcane (IS), intercropping peanut 1 (IP1) and intercropping peanut 2 (IP2) significantly increased by 26.7%, 16.0% and 65.3% than monocropping sugarcane, respectively. IP2 showed a significantly higher available phosphorus content than IP1. The available nitrogen content was significantly increased in the intercropping treatments, except in IS, when compared with MS; it increased by 7.5% and 20.1% in IP1 and IP2, respectively, while it decreased by 7.3% in IS. In IS, IP1 and IP2, the available potassium contents were all significantly lower in the root zone soil than in MS, decreasing by 7.4%, 14.5% and 7.4%, respectively. IP2 showed a significantly higher available potassium content than IP1.

Intercropping significantly increased the total nitrogen content, as shown by comparing the MS and other treatments. The total nitrogen content of IS, IP1 and IP2 increased 18.5%, 16.5% and 45.5%, respectively, compared to MS, and IP2 showed a significantly higher total nitrogen content compared to IP1. The total phosphorus content showed a decreasing trend in the intercropping treatments. This content decreased by 12.5%, 35.8% and 45.7% in IS, IP1 and IP2, respectively, although the decrease was not significant in IS.

Tang et al. (2021), *PeerJ*, DOI 10.7717/peerj.10880

**Table 1** **Basic soil physicochemical properties of MS, IS, IP1 and IP2 in root zone soils.** +/- indicated standard error. The combinations of letters a, b, c, and d beside the values in the table indicate statistically significant groups. Each experimental group contained 3 field replicates for each of the four treatments for a total of n=12 altogether. Different letters in the same column represent significant differences.

| Treatments | Available nitrogen (mg·kg⁻¹) | Available phosphorus (mg·kg⁻¹) | Available potassium (mg·kg⁻¹) | Total nitrogen (g·kg⁻¹) | Total phosphorus (g·kg⁻¹) | Total potassium (g·kg⁻¹) | Organic matter (g·kg⁻¹) | pH value | Water content (%) |
|---|---|---|---|---|---|---|---|---|---|
| MS | $92.867 \pm 2.458c$ | $91.607 \pm 1.528d$ | $135.333 \pm 3.786a$ | $0.709 \pm 0.045b$ | $0.862 \pm 0.152a$ | $6.167 \pm 0.804c$ | $16.385 \pm 0.455b$ | $6.903 \pm 0.021c$ | $11.667 \pm 0.882b$ |
| IS | $86.100 \pm 2.524d$ | $116.107 \pm 1.041b$ | $125.333 \pm 3.215b$ | $0.840 \pm 0.058b$ | $0.754 \pm 0.245ab$ | $7.500 \pm 0.250b$ | $16.233 \pm 0.573b$ | $7.010 \pm 0.010b$ | $14.778 \pm 0.839a$ |
| IP1 | $99.867 \pm 2.650b$ | $106.274 \pm 1.258c$ | $115.667 \pm 2.517c$ | $0.826 \pm 0.037b$ | $0.553 \pm 0.048ab$ | $10.833 \pm 0.629a$ | $24.653 \pm 1.481a$ | $7.113 \pm 0.025a$ | $10.889 \pm 1.018b$ |
| IP2 | $111.533 \pm 0.808a$ | $151.440 \pm 0.500a$ | $125.333 \pm 2.309b$ | $1.031 \pm 0.008a$ | $0.468 \pm 0.050b$ | $7.250 \pm 0.250b$ | $25.563 \pm 0.263a$ | $6.927 \pm 0.015a$ | $14.445 \pm 0.385a$ |
| n | 12 | 12 | 12 | 12 | 12 | 12 | 12 | 12 | 12 |
| *P* value | <0.001 | <0.001 | <0.001 | <0.001 | 0.041 | <0.001 | <0.001 | <0.001 | <0.001 |

**Table 2  5 major enzyme activities of MS, IS, IP1 and IP2 in root zone soils.**

| Treatments | Catalase (U·L$^{-1}$) | Urease (U·L$^{-1}$) | Sucrase (U·L$^{-1}$) | Acid phosphatase (U·L$^{-1}$) | Protease (U·L$^{-1}$) |
|---|---|---|---|---|---|
| MS | 43.322 ± 0.573a | 1.098 ± 0.036a | 0.421 ± 0.005a | 2.109 ± 0.071d | 40.579 ± 1.070a |
| IS | 31.875 ± 0.904c | 1.138 ± 0.021a | 0.395 ± 0.001b | 2.274 ± 0.017c | 38.006 ± 5.991ab |
| IP1 | 34.194 ± 0.911b | 1.142 ± 0.019a | 0.378 ± 0.002c | 2.451 ± 0.060b | 33.579 ± 0.943bc |
| IP2 | 31.804 ± 0.960c | 0.904 ± 0.018b | 0.376 ± 0.003c | 2.594 ± 0.065a | 31.006 ± 1.415c |
| n | 12 | 12 | 12 | 12 | 12 |
| P value | 0.000 | 0.000 | 0.000 | 0.000 | 0.008 |

Relative to the MS, the total potassium content was significantly higher in the root zone of IS, IP1 and IP2, with percentage increases of 21.6%, 75.7% and 17.6%, respectively. IP1 showed a significantly higher total potassium content than IP2.

Compared with the MS, a significant increasing trend of organic matter was found in IP1 and IP2, which were increasing by 50.5% and 56.0%. The pH value significantly increased in IS and IP1 by 1.6% and 3.0%, respectively, compared to MS, and there was no significant difference between MS and IP2. IP1 showed a significantly higher pH value than IP2. We also found that IP2 exhibited significantly higher water content compared to the other treatments, while IS showed the lowest water content by a significant margin. Compared to the MS, the water content increased by 23.8% in IP2, decreased by 6.0% in IP1 and decreased by 59.1% in IS.

## Effects of sugarcane/peanut intercropping on soil enzyme activity

A comparison of enzymes in root zone soil is shown in Table 2. The catalase content was significantly higher in MS, with percentage increases of 26.4%, 21.1% and 26.6%, respectively, than in IS, IP1 and IP2. IP1 showed a significantly higher catalase content than IP2. The urease content showed a significant decrease in IP2 compared to MS, while such differences were not shown in MS, IS and IP1. The sucrase content decreased by 6.2%, 10.2% and 10.6% in the intercropping system compared to the MS, although there was no significant difference among the four types of treatments. Intercropping resulted in a significant increase in the acid phosphatase content relative to MS, and the content in IS, IP1 and IP2 increased by 7.8%, 16.2% and 23.0%, respectively. Compared to the MS, the protease content decreased by 6.3%, 17.2% and 23.6% in IS, IP1 and IP2, respectively, where the decrease was significant except in IS.

## Effects of sugarcane/peanut intercropping on the quantity of microbial communities in the root zone soil

Intercropping affected the diversity of soil microbes in root zone soils (Table 3). The number of bacteria in IS, IP1 and IP2 significantly increased by 22.6%, 80.7% and 6.5%, respectively, relative to MS, and the number in IP1 was significantly higher than that in IP2. Compared with the MS, there was a significantly higher number of fungi in the root zone soils of IP2, of which the number increased by 125%. IP2 showed a significantly higher number of fungi than IP2. Relative to the MS, a slight increasing trend of the number of actinomycetes was found in IS, and a slight decreasing trend was found in IP1 and

**Table 3 Microbial quantity and chemical properties of MS, IS, IP1 and IP2 in root zone soils.**

| Treatments | Bacteria ($10^5$ g$^{-1}$) | Fungi ($10^2$ g$^{-1}$) | Actinomycetes ($10^5$ g$^{-1}$) | Microbial biomass nitrogen (mg·kg$^{-1}$) | Microbial biomass carbon (mg·kg$^{-1}$) | Microbial biomass phosphorus (mg·kg$^{-1}$) |
|---|---|---|---|---|---|---|
| MS | 10.333 ± 0.577c | 4.000 ± 1.000bc | 20.000 ± 3.606a | 45.372 ± 2.021b | 489.694 ± 5.658a | 8.613 ± 0.194d |
| IS | 12.667 ± 0.577b | 3.000 ± 1.000c | 20.667 ± 2.517a | 57.038 ± 2.021a | 445.500 ± 4.451c | 11.583 ± 0.115b |
| IP1 | 18.667 ± 0.577a | 4.000 ± 1.000bc | 16.667 ± 1.155a | 53.538 ± 2.021a | 418.414 ± 8.642d | 10.389 ± 0.161c |
| IP2 | 11.000 ± 1.000c | 9.000 ± 1.000a | 19.000 ± 1.000a | 53.538 ± 2.021a | 465.458 ± 3.266b | 15.853 ± 0.060a |
| n | 12 | 12 | 12 | 12 | 12 | 12 |
| P value | 0.000 | 0.000 | 0.229 | 0.072 | 0.041 | 0.046 |

IP2, although both the increase and decrease were not significant. The biomass nitrogen content increased significantly by 25.7%, 18.0% and 18.0% in IS, IP1 and IP2, respectively, compared to MS, and the biomass carbon content decreased significantly in intercropping treatments. This content decreased by 9.0%, 14.6% and 5.0% in IS, IP1 and IP2 compared to the MS. The biomass phosphorus content increased significantly by 34.5%, 20.6% and 84.1% in IS, IP1 and IP2 compared to the MS, and this difference was also significant between the two treatments.

## Abundance of metabolic pathways in sugarcane/peanut intercropping

The relationships of 32 different metabolic pathways were analyzed using the KEGG database (Fig. 2). According to the results, we analyzed abundances of pathways related to different metabolisms. 11 of these pathways were related to carbohydrate metabolism and 7 to amino acid metabolism, of which abundances in different treatments were various. The rest of pathways includes lipid metabolism, nucleotide metabolism and biosynthesis of other metabolites which also showed differences between treatments.

We found that the pathways involved in carbohydrate metabolism and amino acid metabolism were more abundant than other metabolic pathways, and the abundances in the IP2 treatment were generally higher than those in the other treatments (Figs. 2 and 3). Carbohydrate metabolism pathways include purine metabolism, glycolysis/gluconeogenesis, and pyruvate metabolism.

## Abundance of genes involved in N cycling, P cycling and plant degradation

According to the analysis of gene abundances (Fig. 4), *glnA* (K01915), *GLUD1_2* (K00261), and *nirK* (K00368) were the most abundant genes for N reactions. The relative abundance of *glnA* was 12.6% higher in the IP1 treatments, and the relative abundance of *GLUD1_2* was 25.5% higher in the IP2 treatment compared to the monocropping treatment. The abundance of *nirK* was 12.0% higher in IS than MS, and *ncd2* (K00459) was more abundant in IP1 than MS in Proteobacteria (Fig. 5).

For P cycling, the abundances of *phoR* (K07636), *phoB* (K07657) and *phoB1* (K07658) were higher than those of other genes (Fig. 4). Moreover, the abundances of those genes were also higher in the intercropping treatments than in the monocropping treatment. The abundances of *phoR* and *phoB1* were 13.8% and 3.2% higher in IP2, and the abundance

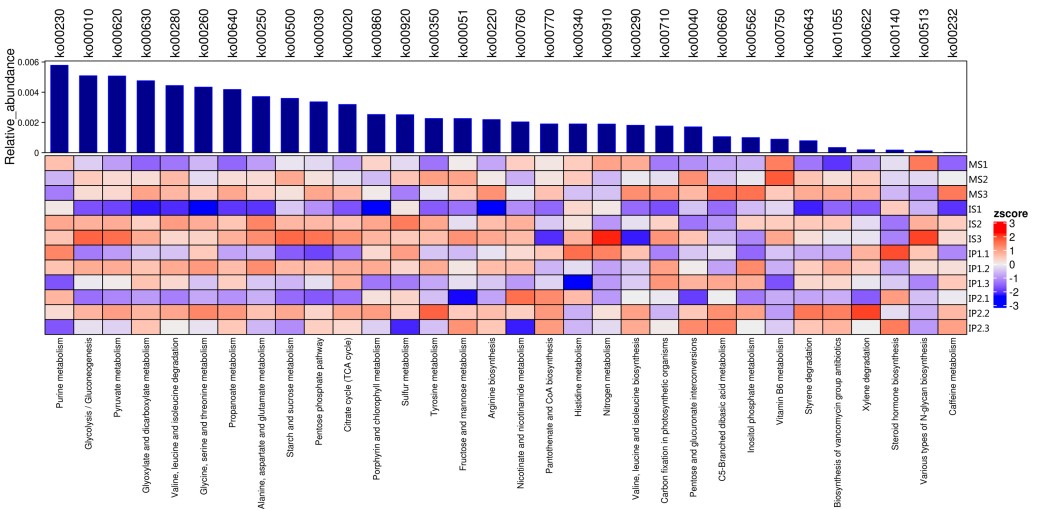

**Figure 2   Metabolic pathways for N and P metabolism and other types of metabolism related to organic matter turnover in the microenvironment.** Relative abundances of each type of metabolism. Blue bars represent the total relative abundances in four treatments and the heatmap indicates the relative abundance in each treatment.

of *phoB* was 4.2% higher in IS than in MS. More genes, including *phoA* (K01077), *mmsA* (K00140) and *TPI* (K01803), were more abundant in IP1 than in MS.

For plant degradation, the abundances of *bgIX* (K05349), *PRDX2_4* (K03386) and *GAPDH* (K00134) were higher than those of other genes. Among these genes, *bgIX* and *PRDX2_4* were 21.9% and 10.5% more abundant in IS than MS, while *GAPDH* was more abundant in MS than in intercropping treatments (Figs. 4 and 5). Furthermore, in the dominant phylum Acidobacteria, *yvak* (K03928) and *xynB* (K01198) were more abundant in IS than MS, and *katG* (K03782) was more abundant in IP1 than MS (Fig. 5).

In network analysis between genes related to N and P cycling and plant polymer degradation, *PRDX2_4* and *nirK*, as shown in the former results of higher abundances in intercropping, also had high betweenness centrality scores (Fig. 6). Genes with high scores generally showed higher abundances in intercropping than monocropping.

# DISCUSSION

## Sugarcane/peanut intercropping system changed the physicochemical properties of root zone soils

Previous studies have shown that intercropping systems have an important impact on the content of various nutrients (*Liu et al., 2019a*; *Wang et al., 2015*). Our study indicated that the content of available nitrogen, available phosphorus, total nitrogen, total potassium, organic matter and pH value in root zone soil increased, and the content of available potassium, total phosphorus and water decreased, in intercropping treatments compared to the monocropping treatments.

Studies have shown that the content of available nitrogen and phosphorus increased in cassava/peanut intercropping (*Li et al., 2012*), and the content of total nitrogen, phosphorus

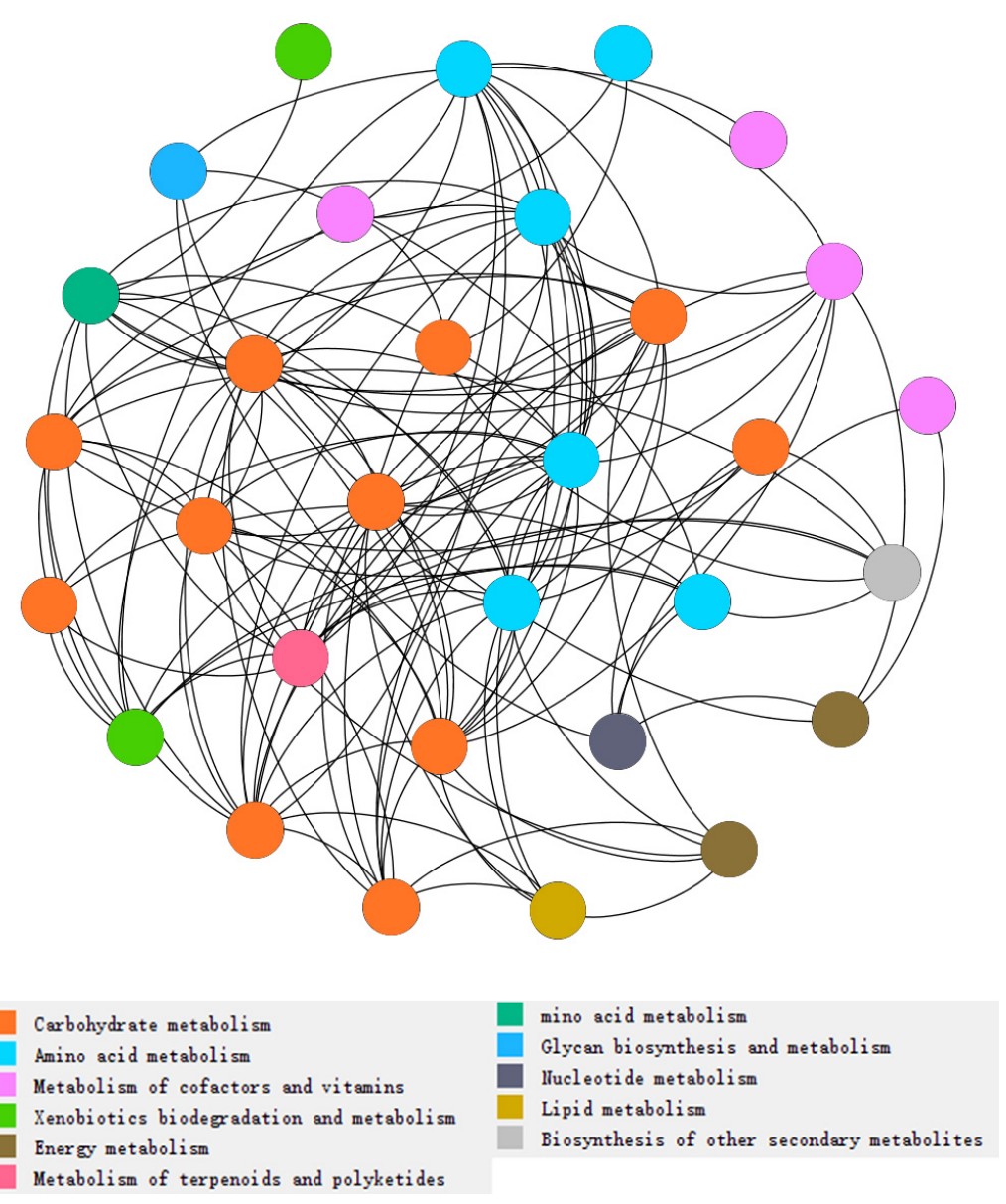

Carbohydrate metabolism
Amino acid metabolism
Metabolism of cofactors and vitamins
Xenobiotics biodegradation and metabolism
Energy metabolism
Metabolism of terpenoids and polyketides

mino acid metabolism
Glycan biosynthesis and metabolism
Nucleotide metabolism
Lipid metabolism
Biosynthesis of other secondary metabolites

**Figure 3  Network analysis of types of metabolism involved with N and P cycling and organic matter turnover in intercropping treatments.** Each color indicated a particular metabolism, and the number of dots indicated abundances of metabolism in intercropping treatments. More dots in same color indicates more abundances of a kind of metabolism. Each dot indicates a pathway participated in metabolism.

and potassium also increased according to other researchers (*Peng et al., 2014*). However, the content of total nitrogen decreased in milk vetch/rape intercropping (*Zhou et al., 2019*), while the content of total nitrogen, as well as the available potassium and phosphorus, increased in legume/tomato intercropping (*Dai et al., 2015*), which indicated that nutrients varied greatly in root zone soil due to the different intercropped crops.

The content of nutrients in root zone soil is related to the microbe communities and their biological activities (*Solanki et al., 2019*). Due to the function of rhizobia, peanut
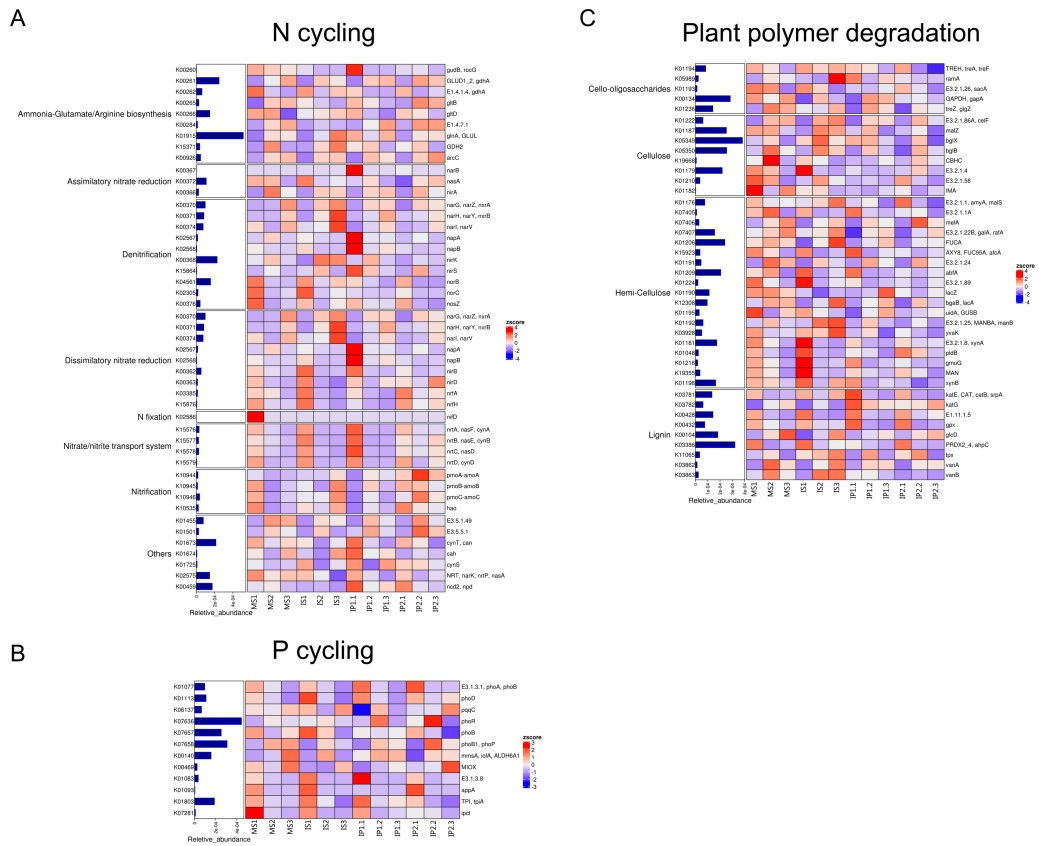

**Figure 4  Relative abundances of genes related to N and P cycling and plant polymer degradation.**
Bars represent the total relative abundances in the four treatments, and the heatmap indicates the relative abundance in each treatment (nonparametric Kruskal-Wallis test).

fixes nitrogen, for which sugarcane has a higher demand. When sugarcane is intercropped with peanut, it may accelerate peanut nitrogen fixation, similar to the situation in the sugarcane/soybean intercropping system (*Li et al., 2012*). Studies have shown that rhizobia accelerate the nutrient absorption of legumes and further increase yield (*Bogino et al., 2011*; *Tian et al., 2019*). Peanut secretes protons and organic acids to activate insoluble inorganic phosphorus (*Lin et al., 2018*; *Liu et al., 2019c*), and the related microbes in soil increase, which enhances the proportion of nutrients and promotes the growth of plants (*Darch et al., 2018*; *Solanki et al., 2018*; *Tang et al., 2016b*). These results suggested that intercropped sugarcane and peanut have some advantages in terms of growth and yield.

The study suggested that the content of catalase, sucrase and protease decreased significantly in intercropping treatments compared to the monocropping treatment. The content of acid phosphatase increased in intercropping treatments, which was also observed in legume/tomato (*Dai et al., 2015*), maize/peanut and maize/soybean (*Zhang et al., 2012a*) intercropping systems. The content of sucrase and urease increased in these systems. However, the content of urease showed no significant difference when all treatments were combined in sugarcane/peanut intercropping.

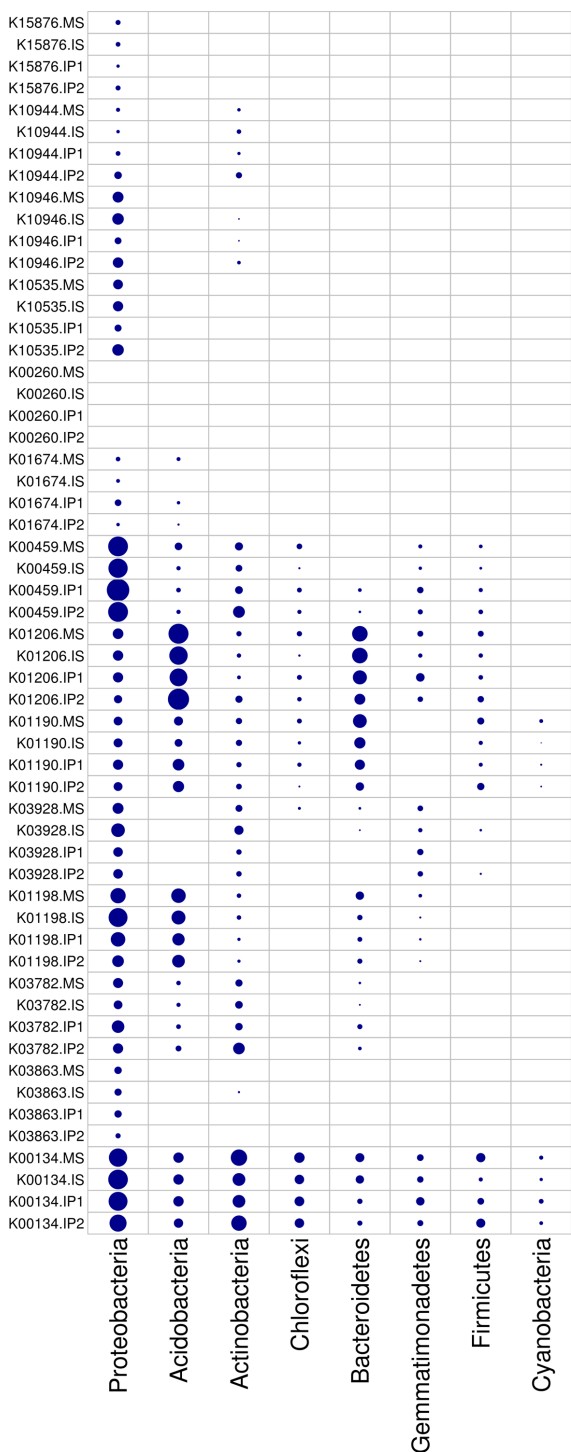

**Figure 5** **Relative abundances of genes in relevant phyla involved in N and P reactions and plant degradation.** The size of the nodes are related to the abundances (larger nodes denote higher abundances). The gene codes are same as in Fig. 4.

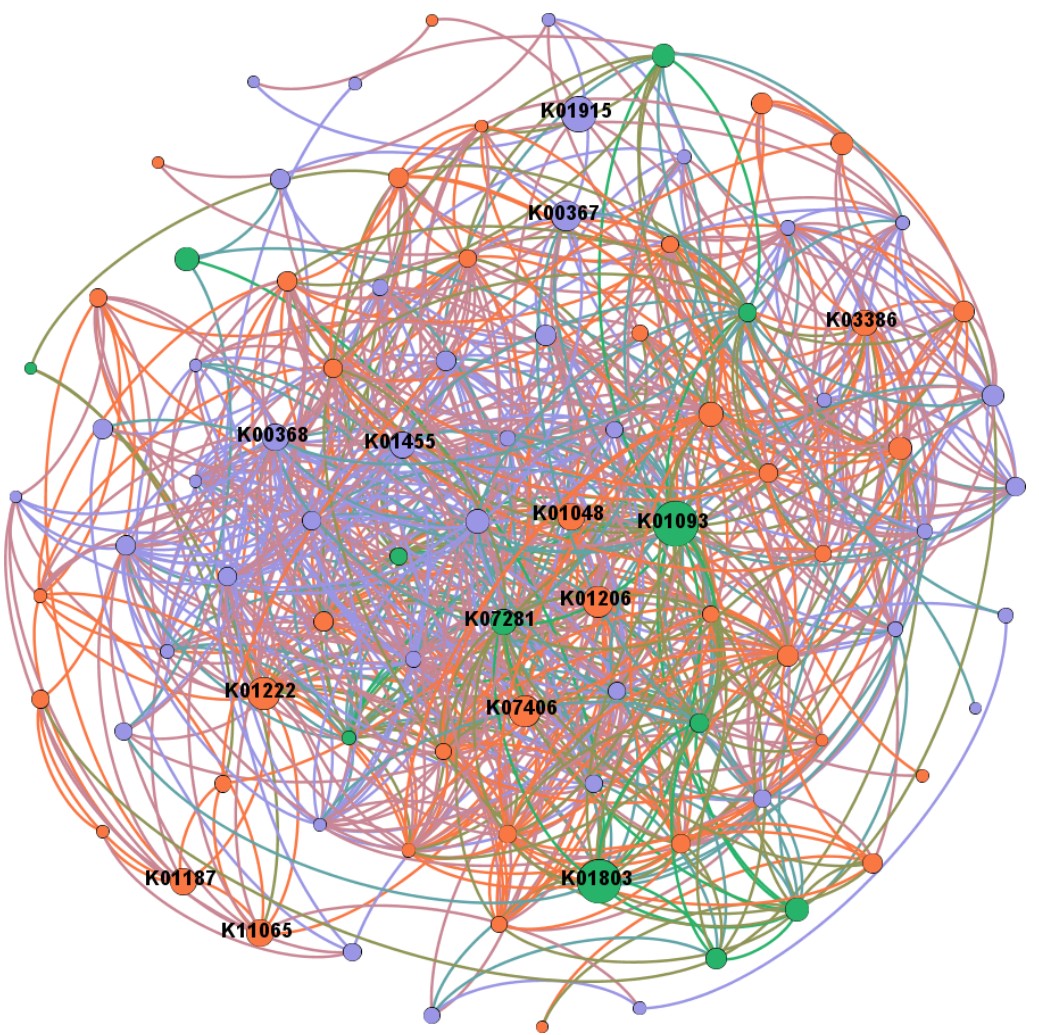

**Figure 6  Network analysis between genes related to N and P cycling and plant polymer degradation.**
Red lines represent significant positive ($p < 0.05$) linear relationships, and blue lines represent negative
($p < 0.05$) linear relationships. Purple nodes are related to genes involved in N reactions. Green nodes are
related to genes involved in P reactions. Yellow nodes are related to genes involved in plant degradation.
The size of the nodes is related to the betweenness centrality scores (larger nodes denote higher between-
ness centrality scores).

In similar studies working on sugarcane/peanut intercropping, researchers found the
content of urease, acid phosphatase and catalase increased in soil (*Chen et al., 2019*). These
differences between their and our results may have resulted from the soil conditions,
species of sugarcane and peanut, fertilizer application, climates and other factors, as
we conjectured. The content of nitrogenase increased significantly in maize/soybean
intercropping, as well as in sugarcane/legume intercropping, thereby influencing the soil
properties and enhancing the diversity of diazotrophic bacteria (*Solanki et al., 2018*; *Solanki
et al., 2016*; *Zhang et al., 2019a*).

Enzymes participate in various chemical cycling reactions related to the growth of plants and have important functions in the soil environment. Studies have shown that enzyme activity is closely correlated with soil chemical properties and microbe activity (*Solanki et al., 2019*; *Wang et al., 2015*). The number of actinomycetes and bacteria significantly affect sucrase, while the number of fungi affects urease and acid phosphatase (*Hu et al., 2002*). Microbial activity connected to metabolic processes results in changes in enzymes and nutrients, which supports the growth of microbial communities. According to the results, the number of bacteria in IS, IP1 and IP2 significantly increased by 22.6%, 80.7% and 6.5%, respectively, relative to MS. Increase of bacteria caused increase of activities of genes, which contributed to higher level of organic matter turnover and enhanced metabolism in root zone soil. In peanut/sugarcane intercropping system, we consider it as an improvement of soil environment and speculate that it would be benificial to the growth of both peanut and sugarcane.

Changes and differences in physichemical properties suggested by our studies between monocropping and intercropping in sugarcane and peanut may be derived from their root interaction. Root interaction plays an important role in intercropping systems. There are competition, as well as promoting, effects in root interactions, especially outstanding in the environment lacking resources (*An, Feng & Zhang, 2017*). The change in the soil environment in intercropping systems will affect the species and structure of microbial communities in soil, enhancing the promoting effect on the absorption of nutrients in roots. As studies have shown, this effect promotes the absorption of nitrogen and phosphorus (*Ling et al., 2018*; *Zhang et al., 2016*), optimizes the difference in water content (*Wang, 2018*) and affects nitrogen fixation (*Regehr et al., 2015*; *Zhao et al., 2020*). Roots grow in different morphologies in different intercropping systems due to the response to identities of neighbors, and different root exudates have an important impact on the growth of plants by affecting the soil environment (*Zhang, 2018*). In sugarcane/peanut intercropping, different demands for nutrients may lead to a more reasonable distribution and higher absorption of nutrients in root zone soil accelerated by the promoting effect of root reactions through dynamic changes in the soil environment.

## Improved root zone soil physicochemical properties were related to the bacterial community in sugarcane/peanut intercropping systems

Relative to the MS, the content of biomass nitrogen and phosphorus increased significantly, and the content of biomass carbon decreased in the intercropping treatments. The content of bacteria and fungi also showed significant differences among these treatments, which indicated that intercropping had an important impact on the structure of microbe communities. The number of bacteria and fungi increased, as was observed by other researchers (*Chen et al., 2019*).

Intercropping significantly affected the diversity of microbes and the proportion of bacteria and fungi. In the cassava/peanut intercropping system, the specific value of bacteria and fungi (B/F) increased first and then decreased with the prolongation of the growth period, which is conductive to the transformation of rhizosphere soil turned into bacterial type (*Xu et al., 2016*). Many studies have shown that the number of microbes,

such as bacteria, fungi and actinomycetes, increased significantly in sugarcane/legume intercropping systems (*Li et al., 2012*; *Solanki et al., 2019*; *Solanki et al., 2018*; *Solanki et al., 2016*). A similar situation occurred in other intercropping systems, including maize (*Chen et al., 2018a*; *Zhang et al., 2012a*) and wheat (*Dong et al., 2013*). However, in the intercropping system of Chinese milk vetch and rape, the content of soil microbial communities decreased (*Zhou et al., 2019*). In cereal/legume intercropping, a significant effect only occurred under high phosphorus levels on the microbial proportion in root zone soil (*Tang et al., 2016b*). These differences demonstrated that microbe quantity and activity were correlated with the species of intercropped crops, intercropping modes, fertilization, soil conditions and other impact factors.

## Metagenomic data analysis

In this study, we selected soil samples from different areas of intercropping crops for metagenomic sequencing. The abundance of such genes as *glnA*, *GLUD1_2* and *gltD* involved in N cycling, including ammonia-glutamate/arginine biosynthesis, was generally higher in intercropping treatments than in monocropping treatments. These genes contributed to the significant abundance of carbohydrate and amino acid metabolism, which was in accordance with the results that metabolism related to these genes was more active in the intercropping system (Fig. 2). Glutamine synthetase, which is encoded by *glnA*, is an essential enzyme in ammonium assimilation and glutamine biosynthesis and plays an important role in nitrogen and carbon metabolism (*Rodriguez-Herrero et al., 2020*; *Xiao et al., 2018*). Glutamate dehydrogenase encoded by *GLUD1* is a key enzyme in glutaminolysis, which converts glutamate to $\alpha$-for entering the TCA cycle (*Craze et al., 2019*). Enzymes encoded by *giltD* participate in the synthesis and degradation of NADPH, functioning in the primary metabolic pathway. The *nirK* and *nirS* genes are important biomarkers for denitrifying microorganisms (*Wang et al., 2020*), and they showed more abundance in IS and IP1 than in MS. Moreover, arginine is also degraded by microbes through many different metabolic pathways, and the difference between treatments may indicate higher activity of microbes in intercropping systems. These results suggested that intercropping may affect the structure and quantities of microbial communities by mediating nitrogen and carbon metabolism, similar to the results of higher available nitrogen and total nitrogen in intercropping obtained by the former study.

*PhoR/PhoB* is involved in the expression of genes related to the acquisition of phosphate and its derivatives (*Santos-Beneit, 2015*), and it showed more abundance in IP2 than MS, which indicated that phosphorus metabolism was more active in the intercropping system. Peanut secretes protons and organic acids to activate insoluble inorganic phosphorus (*Solanki et al., 2018*), which may enhance the nutrient absorption of sugarcane and improve the chemical composition as well as the pH value of the soil environment when peanut is intercropped with sugarcane. The peroxiredoxin (PRDX) gene family is an important conserved antioxidant protein that reduces the number of peroxides in cells through cysteine and thiol electron donors (*Lin et al., 2013*). GAPDH is a key enzyme involved in glycolysis. The gene encoded GAPDH is more abundant in MS than intercropping

treatments, which may indicate that plants need more energy to maintain their own growth in monocropping.

According to the results, we found that pathways involved in carbohydrate metabolism and amino acid metabolism were more abundant than other metabolic pathways in the intercropping system (Figs. 2 and 3). We speculate that it is associated with increase of bacteria as former results mentioned. As gene analysis showed, related genes involved in N cycling, P cycling and organic matter turnover vary significantly between intercropping and monocropping treatments (Figs. 4 and 5), contributing to changes in metabolic pathways and more portions of the soil environment. Act as decomposers in ecosystem, microbes play an important role in anabolism and catabolism. When microbes increased, the activities of genes related to metabolism increased, subsequently leading to more active N/P cycling and organic matter turnover. The synthesis and degradation of carbohydrates is the basis of the growth and fruit maturation of plants and the basic substances essential to the life of microbes. This process contributed to the growth of plants and finally reflected in yield of crops.

According to the results of the correlation analysis (Fig. 7), *TREH* (K01194) and *katE* (K03781) are significantly related to phosphorus metabolism, showing positive effects on available phosphorus, acid phosphatase and microbial biomass phosphorus. Trehalase (encoded by *TREH*) is a glucosidase that hydrolyzes a trehalose molecule into two glucose molecules (*Tang et al., 2016a*), and the *katG* protein encoded by *katG* is a hydrogen peroxidase (*Rong, Lv & Sun, 2011*). In addition, these proteins are more abundant in IS and IP1 than MS (Fig. 4), which means that the intercropping system may affect the phosphorus content and activity of enzymes in root zone soil by accelerating phosphorus-related genes, such as *TREH* and *katE*. This finding was consistent with the results that the content of available phosphorus, microbial biomass phosphorus and acid phosphatase all significantly increased in intercropping compared to monocropping. *gudB* (K00260) was significantly related to the content of available nitrogen and available potassium, and its abundance was higher in IP1 than in MS. The content of available nitrogen and total nitrogen increased in intercropping, which may be caused by the different activities of genes related to nitrogen metabolism.

Intercropping system have shown great importance in agronomy and ecology. Our results help to elucidate the potential responses of genes involved in N and P reactions in peanut/sugarcane intercropping systems. Using metagenome sequencing, we obtained new insights into the mechanisms responsible for interaction in soil environment of peanut-sugarcane intercropping system. Due to the relevance of different metabolic pathways, the intercropping system influenced the abundances of genes involved in various metabolisms and improved the soil environment of root zone soil by mediating the activities of enzymes and microbes (Fig. 8). These finally increase the nutrients in root zone soil which is beneficial to the growth and development of crops.

## CONCLUSIONS

As studies have shown, sugarcane/peanut intercropping significantly affects root zone soil physicochemical properties, enzyme activities and microbial community quantities.

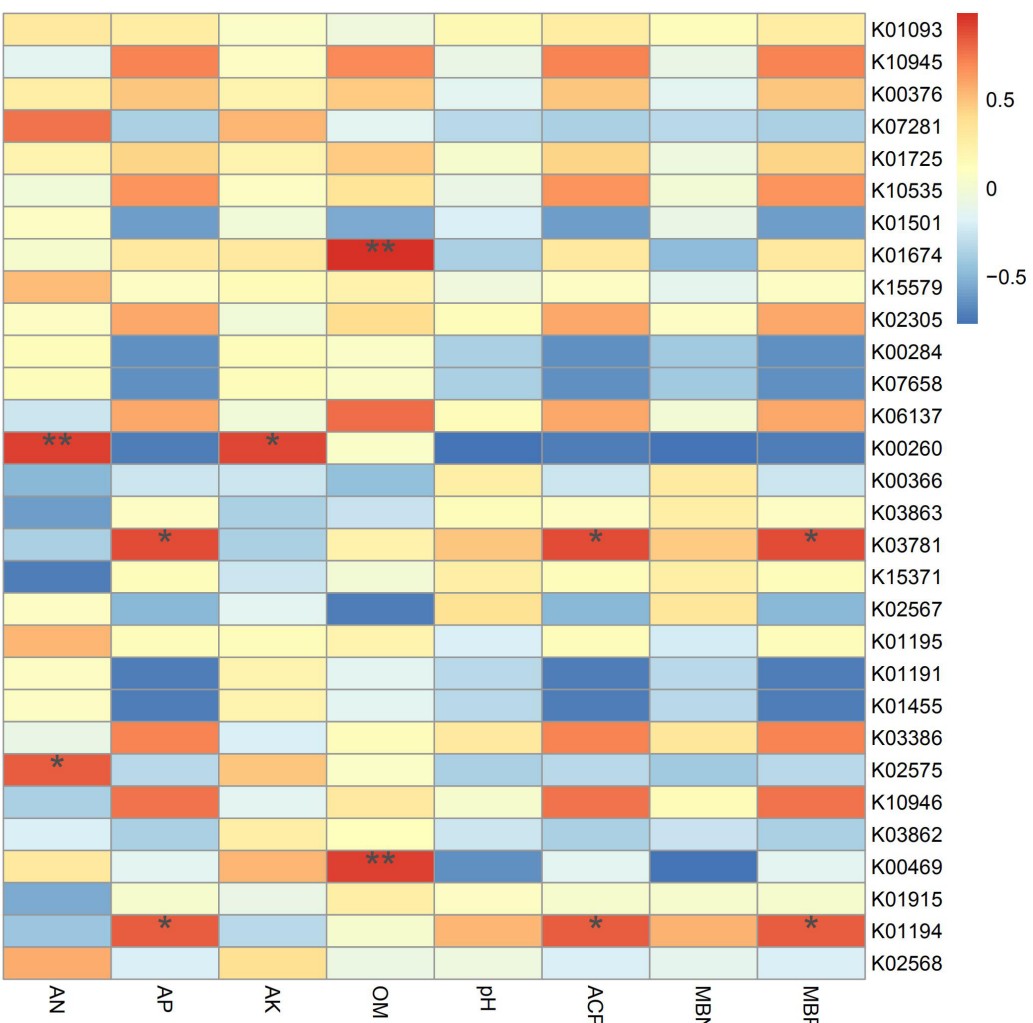

**Figure 7   Spearman's correlation coefficients between key genes in networks and soil properties in different treatments.** Single asterisks and double asterisks indicate $p < 0.05$ and $p < 0.01$, respectively.

Metagenomic analysis suggested that the relative abundances of genes related to N cycling (*glnA*, *GLUD1_2,* and *nirK*), P cycling (*phoR* and *phoB*) and organic matter turnover (*PRDX2_4*) were higher in the soil of intercropping treatments. Genes significantly related to phosphorus metabolism (*TREH*, *katE*, and *gudB*) were more abundant in intercropping than in monocropping. The intercropping system changed chemical properties by regulating genes involved in N cycling, P cycling and organic matter turnover and then improved the soil environment (Fig. 8). Our results provide a theoretical basis for the basic mechanism of the soil environment composed of such elements as nutrients, enzymes, and microbes. Nutrients, enzymes and microbes work together and reach a dynamic balance responsible for the positive or negative effects on the growth of plants, which elucidates the importance and basic reaction mechanism of the soil environment. Further research at the

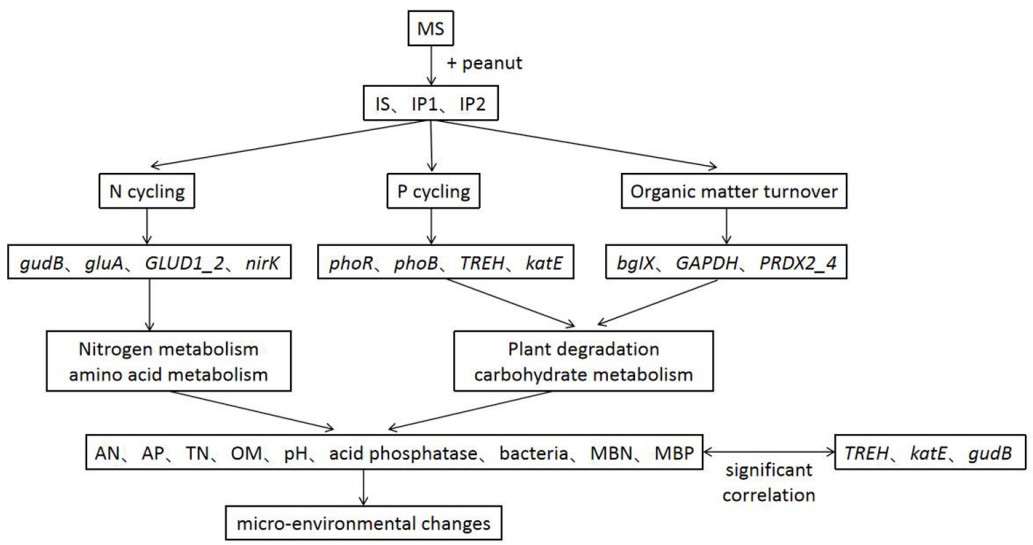

**Figure 8** A conceptual model of intercropping affecting genes, pathways and physichemical properties in root zone soil.

hereditary and molecular levels is needed to elucidate the specific mechanism governing sugarcane/peanut intercropping systems.

## ACKNOWLEDGEMENTS

We thank BGI Genomics Co., Ltd., (Wuhan, Hubei, China) and Pfomic Info Co., Ltd., (Nanning, China) for metagenome sequencing and analysis, and we thank Yong Xu for helping with the graph drawing.

### Funding

These works were supported by the National Natural Science Foundation Project (31660371), the Technology System for Modern Agriculture (CARS-13-Southern China), the Guangxi Natural Science Foundation Project (2017GXNSFAA198144) and the Scientific Project from the Guangxi Academy of Agricultural Sciences (2017JZ12). The funders had no role in study design, data collection and analysis, decision to publish, or preparation of the manuscript.

### Grant Disclosures

The following grant information was disclosed by the authors:
National Natural Science Foundation Project: 31660371.
Technology System for Modern Agriculture.
Guangxi Natural Science Foundation Project: 2017GXNSFAA198144.
Scientific Project from the Guangxi Academy of Agricultural Sciences: 2017JZ12.

## Competing Interests

The authors declare there are no competing interests.

## Author Contributions

- Xiumei Tang conceived and designed the experiments, performed the experiments, prepared figures and/or tables, authored or reviewed drafts of the paper, and approved the final draft.
- Yixin Zhang conceived and designed the experiments, performed the experiments, analyzed the data, prepared figures and/or tables, authored or reviewed drafts of the paper, and approved the final draft.
- Jing Jiang performed the experiments, authored or reviewed drafts of the paper, and approved the final draft.
- Xiuzhen Meng performed the experiments, prepared figures and/or tables, and approved the final draft.
- Zhipeng Huang conceived and designed the experiments, prepared figures and/or tables, authored or reviewed drafts of the paper, and approved the final draft.
- Haining Wu performed the experiments, prepared figures and/or tables, and approved the final draft.
- Liangqiong He conceived and designed the experiments, performed the experiments, prepared figures and/or tables, and approved the final draft.
- Faqian Xiong and Zhuqiang Han conceived and designed the experiments, authored or reviewed drafts of the paper, and approved the final draft.
- Jing Liu conceived and designed the experiments, performed the experiments, authored or reviewed drafts of the paper, and approved the final draft.
- Ruichun Zhong performed the experiments, prepared figures and/or tables, authored or reviewed drafts of the paper, and approved the final draft.
- Ronghua Tang conceived and designed the experiments, prepared figures and/or tables, authored or reviewed drafts of the paper, and approved the final draft.

## DNA Deposition

The following information was supplied regarding the deposition of DNA sequences:

All of the sequence data are available in the NCBI Sequence Read Archive (SRA) database: SRP267937.

## Data Availability

All of the sequence data are available in the NCBI Sequence Read Archive (SRA): SRP267937 (SAMN15324604, SAMN15324605, SAMN15324606, SAMN15324607, SAMN15324608, SAMN15324609, SAMN15324610, SAMN15324611, SAMN15324612, SAMN15324613, SAMN15324614, and SAMN15324615).

Raw data are available in the Supplemental Files.

## Supplemental Information

Supplemental information for this article can be found online at http://dx.doi.org/10.7717/peerj.10880#supplemental-information.

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
