# Peer review of "Sugarcane/peanut intercropping system improves physicochemical properties by changing N and P cycling and organic matter turnover in root zone soil"

_PeerJ, doi:10.7717/peerj.10880_

## Round 0.1 · original submission · Major Revisions

The manuscript has been carefully reviewed by two reviewers and both recommended a major revision which I concur. Please revise it accordingly and resubmit.

Reviewer 1 ·

Basic reporting

This manuscript compares soil nutrients, enzymes, and microbial dynamics in the sugarcane/peanut intercropping system. The authors analyzed soil nutrients as well as metagenomes to understand nitrogen and phosphorus cycling in the intercropped systems. The work adds new knowledge on the microbial control of N and P cycling. However, there are some issues that raise a question about the work.

Experimental design

The treatment design and sampling plan do not support the title nor objective of the research. It is basically a soil sampling in three rows of intercropping and analysis for differences in root zone soil. Also, it is not clear how long the experimental plots have been established. In materials and methods, authors mention peanut planted on March 10, 2018-2019. Is it established in 2018? If this is single-season work, the discussion and conclusion are too strong and it does not make sense. There is no baseline soil condition and authors did not provide the detail of the experimental field/soil variability. Peanut planted on March 10 and July 25th sampling authors sees an increase in SOC by >8 g/kg, this is unreasonably high, some sampling or experimental biases could have affected the results.

Validity of the findings

This is a single season work. One time sampling in long-term experiments (several years of treatment application) is common but the single-season work may not give the complete picture on how intercropping work, Figure 1 does not provide a complete picture of the experimental design, it shows they are basically collecting samples from three different rows and tested it for different soil properties. There is no baseline soil condition and authors did not provide the detail of the experimental field/soil variability. Peanut planted on March 10, soil sampled on July 25th, and analysis showed an increase in SOC by >8 g/kg. This does not make sense.

Additional comments

This manuscript compares soil nutrients, enzymes, and microbial dynamics in the sugarcane/peanut intercropping system. The authors analyzed soil nutrients as well as metagenomes to understand nitrogen and phosphorus cycling in the intercropped systems. The work adds new knowledge on the microbial control of N and P cycling. However, there are some issues that raise the question on the hypothesis of this project, design and plan of the experiment.
The treatment design and sampling plan do not support the title nor objective of the research. It is basically a soil sampling in three rows of intercropping and analysis for differences in root zone soil. Also, it is not clear how long the experimental plots have been established. In materials and methods, authors mention peanut planted on March 10, 2018-2019. Is it established in 2018? If this is single-season work, the discussion and conclusion are too strong and it does not make sense. There is no baseline soil condition and authors did not provide the detail of the experimental field/soil variability. Peanut planted on March 10 and July 25th sampling authors sees an increase in SOC by >8 g/kg. However, proportionate changes in other soil properties are not observed.
What was the cultivation, fertility, and other management details for each crop? How many years?
How data on soil physicochemical properties were analyzed? Please provide the detail in the statistical analysis section.
Was metagenomics analysis also done in replicated samples? How the sequence data were analyzed (if replicated). The materials and method section needs a drastic revision with so many changes.

L59-60: remove the sentence “facilitation occurs in other intercropping …………………. systems”
L72: remove “Hense”
L94: was sugarcane and peanut planted on March 10, 2018, and March 10, 2019? This is confusing. How many years of intercropping? Changes in soil properties just after one season of intercropping does not make sense
L121: potassium dichromate
L184: significantly increased “available p” by – the sentence as is incomplete.
L195: if 45.7% decrease is not significantly different, there is a major flaw in your data or data analysis (provide
L201-202: rewrite the sentence, it’s confusing what comparisons were made.
Discussion:
Most of the discussion is focused on what happened but did not interpret why/how aspects. For example, intercropping increased the nitrogenase enzyme, but why?
How intercropping increases p cycling, please clarify. Provide linkages between genomic analysis you did and respond in nutrients in the soil.
L312: changes and differences in what?

Reviewer 2 ·

Basic reporting

The language is overall understandable, but there is some lack of clarity and unusual phrasing in places.

References seem appropriate, but there are some formatting corrections needed.

Figures and tables are suitable, but several figures lack the field replicates and thereby prevent the reader from seeing the variability within an experimental group.

Raw physico-chemical soil data are shared as a supplemental excel spreadsheet, but the authors need to add the units to each data column in the spreadsheet. Raw sequence data were confirmed available at the Sequence Read Archive, but assembled contigs should be made available as well.

No hypotheses are expressed. The objectives are purely exploratory and could be improved by indicating specific motivations or expectations based upon prior findings in the literature or perhaps from preliminary work within this research group.

Experimental design

The scope is appropriate for PeerJ and is a primary research work.

The research question and motivations for the work could be better stated toward the end of the introduction alongside the objectives.

The data visualization needs further improvement.

There are deficiencies in the methods reporting, in particular citations to the protocols or methods used are needed.

Presently more detail is needed in order to be able to replicate the methods.

Validity of the findings

While there is replication within the study, it is not indicated in several heatmap figures. Significant findings are given in the conclusion section, but need to be expressed quantitatively rather than just in qualitative terms.

The data are provided, and statistical analyses have been conducted. There is a control, sugarcane monocropping group. There is some concern over possible confounding effects of different fertilization regimes between the sugarcane and sugarcane + peanut groups.

As mentioned previously, the objectives/hypotheses and research question need to be more clearly stated and inter-related in the introduction. Clearer linkage that indicates resolution of the research question is needed in the conclusion.

The discussion needs to clearly link leads from the literature to related or contrasting findings in the data from this study.

Additional comments

The researchers compared sugarcane intercropping vs. monocropping to determine whether there was an effect of the cropping on soil fertility parameters including physical and chemical properties and genes responsible for microbial nutrient cycling functions. This is a worthwhile study, but in its present form it requires a number of corrections to ensure that, for example, the data in the figures indicate to the reader the variability within each experimental group. Also, the discussion needs to directly relate findings in the literature to results of this study by citing the values obtained in this study and the supporting figures and tables. This direct linkage is needed throughout the discussion. The writing is understandable, albeit unclear in a few places.

Specific comments:

Abstract background: Although this needs to be kept short, it could perhaps be made more informative--what benefit is there to intercropping these 2 crops in south China? The authors state that it is efficient, but can they be quantitative--for example, what amount of N-fertilization reduction is possible when intercropping the leguminous peanut? Is there greater combined harvest or cumulative economic gain to intercropping?

Abstract results: Can the authors specify by exactly how much the genes and enzyme activities increased?

Abstract conclusion: How do the authors know that it improved the microenvironment? What constitutes an improvement? From an agronomy standpoint, did the soil produce higher yield? If so, is there a way to determine whether the microbial component of soil fertility factors really account for this?

lines 78-81: study objectives are currently stated as purely exploratory. Were there any hypotheses in mind as suggested by prior results in the literature, such as that intercropping would improve specific aspects of the soil in certain ways, or that particular microbial functions would be increased or decreased with intercropping?

line 94: What was the prior cultivation and management history of the site? Briefly state this so as to assure the reader that there are no legacy effects from prior management. Is this really only a field trial measuring the effects of just 1 year of cultivation of these experimental groups? This seems like a short time, but if rhizosphere soil effects are what are being measured, then one could reasonably expect differences to be noted within a single season. If this is the case, it may be worthwhile to state this in the discussion.

lines 102, 103: An aerial photo of the randomized plots with boxes overlaid to show each plot would be useful and provide a clearer picture of the entire field layout. It could be added as an additional panel to figure 1.

lines 103-106: What impact would different fertilization regimes applied to the different experimental treatments have on the results of this study, in particular the N, P, and K measurements?

lines 111, 112: The authors state that both loose soil (do they mean bulk soil here?) and root-adherent (rhizosphere) soil were collected, but were data collected separately for these two soil types? It seems like only rhizosphere soil was analyzed, or were they mixed together? If they were mixed together, then there was likely more influence of the bulk soil than the rhizosphere soil since the bulk soil volume would have been greater. Regardless, the authors need to clarify this in the text.

lines 118-133: The methods section needs citations to methodology articles or handbook protocols for all these methods. Were there any deviations to the protocols? Were there any other missing details of importance like input quantities? How about specifying the equipment models and relevant settings that were used to acquire the measurements?

line 135: Is there need to mention use of special equipment perhaps for a bead beating step not detailed in the manufacturer's protocol? Also, tell whether and how the soil may have been homogenized prior to DNA extraction. Specify the amount of soil used for each extraction.

lines 145, 146: The authors are using a new, less common sequencing platform, and so it would be good to provide a citation indicating that it has been shown to have comparable performance specifications a well-known platform like Illumina.

lines 176-178: Simply give the accession number range since they are consecutive to save space. Write "SRA accessions SAMN15324604 to SAMN15324615".

This reviewer has confirmed that the SRA project is accessible online at the SRA.

But what about the assemblies, can the contigs also be provided online?

lines 137-138: Tell DNA what aspect of DNA quality was confirmed by running on a gel--was it size, fragmentation, relative quantity?

lines 243-264: Give values for the gene abundances.

298-304: The paragraph needs some work to more clearly distinguish what are the findings of other studies vs. the findings of this study.

lines 335, 336: How does the proportion of both bacteria and fungi increase together? It seems that either one or the other should increase in proportion to the other.

lines 374-376: Last sentence--what is the code gene, is that a typo? Also, the linkage between your results of microbial analysis and the plant physiology are unclear. Please reword.

lines 399-404: The last paragraph needs some tweaks to the English and could use overall strengthening.

lines 466-468: It seems most publication titles in the references do not capitalize the first letter of each word in the title, but some such as this one do. Ensure consistent capitalization of titles throughout your reference list. Review all references and correct throughout.

lines 525, 546, 547, 588: Italicize species names throughout the reference list.

lines 594: There is an extra space in "quantity, and" that needs to be deleted.

line 607: Should "ab initio" be italicized?

Fig. 1: The (IP2) abbreviation is needed at the end of the figure description. It is recommended also add wording to indicate that this is a "field plot schematic". It is not clear whether IP1 or IP2 have just 1 or 2 or even 3 rows of peanut within their respective field plots, and the authors should clarify this here or in the materials and methods section, or perhaps both.

Fig. 2: Use punctuation (a period or full stop) at the end of each sentence in the figure legends. This figure needs to show the data for each of the 3 replicates in all 4 treatments so that the reader has a clear, visual indication of how much variation occurs within treatment; that is, there should be 12 rows in this figure.

Fig. 3: It is unclear how to interpret this network. The data upon which the network is constructed should be indicated in the legend for the figure. Also, when there are multiple occurrences of a circle of a single color (e.g. orange circles), what does each circle indicate, separate samples perhaps? This needs to be indicated in the figure legend.

Fig. 4: Replicates should be indicated as stated in Fig. 2. The legend should detail precisely the normalized unit used for the read counts. The units for the bar plot are difficult to read at the current figure resolution. Vector versions of the figures should be used instead of raster images.

Fig. 5: A legend for circle size is missing in the figure. Tell what the codes on the left correspond to (i.e. are they KEGG identifiers?). Again, replicates are missing as in Fig. 4 and 2.

Fig. 6: Indicate the underlying data for the figure--is it relative abundances for KEGG groups?

Fig. 8: Indicate the type of model; that is, is there a particular statistical modeling technique applied, or is this a conceptual model? The phrase "micro-environmental changes" is uncommon, and thus it is not entirely clear what is meant in the manuscript by this phrase. Do the authors mean microbiota-induced changes in the environment? Correct this phrasing throughout the manuscript.

Table 1: Specify in the tables what is indicated by +/-, either Standard Error or Standard Deviation. Specify this in the legend or as a footnote in the table. Also specify that the combinations of letters a, b, c, and d beside the values in the table indicate statistically significant groups. Also specify the significance cutoff used. Rather than specify n=12 in each column, specify it just once in the legend for the tables stating that each experimental group contained 3 field replicates for each of the 4 treatments for a total of n=12 altogether. For p-values that are below a threshold and presently listed as 0.000, change to, for example, <0.001.

---

## Round 0.2 · Minor Revisions

Please revise the materials and method section as suggested and resubmit. I still believe that the manuscript will benefit from language editing.

Reviewer 1 ·

Basic reporting

The materials and method section is still not clear. Specifically, the authors mentioned it is two years study but two years of intercropping treatments and sequence of planting and intercrop management have not been provided. Other than that they have addressed all comments in the revised manuscript.

The manuscript will benefit from language editing.

Thank you.

Experimental design

Yes, all issues addressed except timeline of two years of study.

Validity of the findings

It looks reasonable

Additional comments

The materials and method section is still not clear. Specifically, the authors mentioned it is two years study but two years of intercropping treatments and sequence of planting and intercrop management have not been provided. Other than that they have addressed all comments in the revised manuscript.

The manuscript will benefit from language editing.

---

## Round 0.3 · accepted · Accept

Thank you for the revisions.